## [Transparent Peer Review file · Nature Communications]

Orientation-dependent mutual crystalline and amorphous order in a single phase solid

Corresponding Author: Professor Mark Huijben

Version 0:

Reviewer comments:

Reviewer #1

(Remarks to the Author)

General comment:

This paper deals with the elaboration and deep structural analysis of samples made of periodic stacking of niobium and tungsten oxide layers without in plane long range order. The authors describe this 1D ordering without in plane long range order as a new state of condensed matter, which they call "amorphous-crystalline". Such ambition is of course very high and it corresponds to noteworthy results.

Some experimental aspects support the author's assumption. Nevertheless, the manuscript is too hard to read, and figures are often very difficult to understand. The writing of all the paper has to be reconsidered in order to be more readable. One of the aspects that makes reading difficult is the continuous transition, throughout the text, between the orientation of the columns of atoms and the crystallographic orientation of the nanorods. In addition, often the authors refer their scientific comments to supplementary data or notes. This increases the difficulty for the readers. This aspect must be carefully reconsidered case by case.

A large part of the experimental results is strongly related to crystallography. As a result, a stronger full crystallographic notation and description must be used. Due to the fundamental influence of the crystallographic orientation of the substrate surface, all the crystallographic results (in the text and in the figures) have to be presented with respect to the crystallographic axis of these substrates instead of using the q_x , q_y and q_z vector components. These components are related to the measurement conditions, and therefore their meaning often changes, sometimes even in the same figure.

Points to be improved or answered

1- Page 4: according to page 4, the image reported fig2a, is a plane view and in this case (SrTiO₃ (011) substrate) the Nb-W-O nanorods have grown in the plane, along the [100] direction (see Fig. 1g). It seems to mean that the atom columns are along the [011] direction. This must be described in an easier understanding way.

2- On Fig. 2, the images reported in a, b, c and d are corresponding to observation of the columns that have grown on the SrTiO₃(011) substrate. Please, separate this case from that one reported in fig. 2e, that is related to the interface with a SrTiO₃(001) substrate. In the current state, Fig. 2 is really hard to understand. Moreover, the main crystallographic directions must be reported on the images.

3- The contrast observed on Fig. 2c must be discussed. How can a 1D atomic ordering results on such contrast in the image in the TEM? Such image, that is fundamentally an interferences image, comes from a volume of matter. Please give some explanations of the observation in direct space (high resolution TEM image) of a periodic arrangement in transmission mode in such a case as only a 1D periodicity. Moreover, the Fourier transform (Fig. 2d) of this image seems to evidence the presence of regularly spaced points along the straight diffuse lines.

4- Page 10 and 11, Fig.4: 3D RSM are recorded on the sample. According to the size of the x-ray beam, each RSM is due to a large number of columns. How the authors have separated the influence of both the intra-nanorods disorder from the rotational disorder coming to the fact that a large number of nanorods are simultaneously probed?

5- Fig. 4 is very hard to read. As written above, the first improvement is to replace the qx, qy and qz vectors notation by the true crystallographic directions in the reciprocal space. The relationships between the Q-vector components and the crystallographic directions depend on the growth direction, that itself differs when the three types of substrate cutting are considered. In the present form, the notations reported in this figure are too confusing.

Reviewer #2

(Remarks to the Author)

The manuscript entitled "Crystalline or Amorphous? A Matter of Perspective" by Xia et al. demonstrates an interesting inorganic matter which is amorphous in two dimensions but crystalline in the third. The crystalline (including quasicrystal) and amorphous structures are two states for solid matters. They are commonly believed to be mutually exclusive that a state of matter cannot be both crystalline and amorphous at the same time. Usually, a partially crystalline or amorphous matter is just a composite of purely crystalline and amorphous parts. Hence, the matter demonstrated in this manuscript presents experimentally a novel state which is only ordered in certain dimensions.

In fact, such a state of matter has been theoretically proposed as paracrystalline. The term paracrystal refers to a state of matter with less than three-dimensional order, characteristic of a true crystal (Ref: <https://www.merriam-webster.com/dictionary/paracrystal>). Paracrystals are often used in materials science to understand the transition between crystalline and amorphous states. Although the paracrystalline matter has been demonstrated microscopically in many papers (Science 335, 950, (2012); Nature 599, 605 (2021); JAP 90, 4438 (2001)), a direct observation of such matter in bulk is still missing. In this manuscript, we can see the micrometer-scale formation of paracrystalline matter, and the structural characterizations provide compelling evidence in a degree of intermediate order between crystalline and amorphous materials. This manuscript contributes significantly to the existing knowledge on the state of solid matter.

Reviewer has carefully examined the manuscript and supports its publication in Nature Communications. However, it will be shown in the following comments why this matter should be considered as paracrystalline rather than amorphous in one direction.

(1) In Fig. 1, Nb-W-O nanorod films were deposited on crystalline substrates of different orientations. It is a smart approach to demonstrate the difference in structures between different orientations of a matter by showing the various growth morphologies. However, why do these rods all have a diameter around 50 nanometers? If it is truly amorphous in the x-y plane, the diameters of these rods should be infinite or distributed randomly.

(2) What would happen if the films were deposited on an amorphous substrate?

(3) In Fig. 2a and b, the author tries to prove the amorphous structure of the x-y plane by showing its STEM image and the corresponding FFT pattern. However, when the review carefully checks them, some tiny crystalline features can be observed as shown in the following Figures 1R, 2R and 3R. Hence, the x-y plane cannot be understood as fully amorphous but paracrystalline phase.

Fig. 1R. (a) STEM image of Fig. 2a in manuscript and (b) its corresponding halo FFT pattern. If we tuned the intensity of the FFT pattern, some diffraction spots corresponding to crystalline phase can be observed.

Fig. 2R. If we zoom in Figure 2a, a lot of tiny crystals can be observed.

Fig. 3R. Another region where the tiny crystals can be observed. Such regions are distributed over the whole image.

(4) The most important feature of amorphous matter differing from crystalline matter is the glass transition. Can the glass transition be observed in this matter by differential scanning calorimetry? If not, then this matter is paracrystalline rather than amorphous.

By the way, some typos can be seen in the manuscript. "F PDF of Nb-W-O nanorods".

In summary, the state of matter in this manuscript is still unclear. The author needs to verify which kind of structure it is, amorphous or paracrystalline?

Reviewer #3

(Remarks to the Author)

This paper is interesting, though perhaps not as generally applicable as the authors imply in their abstract. The manuscript is well-written, and the characterization work seems solid. However, as I describe below, the authors need to update their knowledge of oxide structures beyond the year 1932 when Zachariasen published his seminal work on glass structures.

My primary issue with this paper is that it's not as original as the author's claims imply. While the atomic resolution characterization in the current work is excellent, I have seen very similar structures (seemingly ordered along one axis) in several previous manuscripts: (none of which are cited by the authors)

- Miran Mozetic, Uros Cvelbar, Mahendra K. Sunkara, and Sreeram Vaddiraju, A Method for the Rapid Synthesis of Large Quantities of Metal Oxide Nanowires at Low Temperatures, Adv. Mater. 17, 2138 (2005).

- Subhra Jana and Robert M. Rioux, Seeded growth induced amorphous to crystalline transformation of niobium oxide nanostructures, Nanoscale, 4, 1782 (2012).

-Sophia B. Betzler, Ai Leen Koh, Bettina V. Lotsch, Robert Sinclair, Christina Scheu, Atomic Resolution Observation of the Oxidation of Niobium Oxide Nanowires: Implications for Renewable Energy Applications, ACS Appl. Nano Mater, 3, 9285 (2020).

Given that the current paper primarily focuses on structure, I expect to see a much deeper dive into other imaging studies on similar nanowires / materials.

In fact, I believe the most similar crystallographic system to the current study isn't mentioned by the authors at all: 2D quasicrystals / (semi)rational approximates in ternary oxides. Figures 3a, b, and especially Figure 3c look like 2D decagonal quasicrystals to me. I suspect there may be additional disorder introduced by substrate and growth effects which may add some randomness to the quasicrystal lattice. For example, in Figure 2a, the (pseudo)coherent domain length appears to be only ~4 nm. Either way, the authors certainly need to explain exactly how their proposed structural models are similar to and distinct from 2D stacked quasicrystals. Here are a few references to get them started (there are many possible references here)

-https://link.springer.com/content/pdf/10.1007/978-3-642-58434-3_3 (textbook chapter)

-Stefan Förster et al., Quasicrystals and their Approximants in 2D Ternary Oxides, *physica status solidi (b)* 257, 1900624 (2020).

-Catalina Ruano-Merchan et al., Stoichiometry-Driven Formation of Two-Dimensional Ternary Oxides: From Quasicrystal Approximants to Honeycomb Lattice Structures, *J. Phys. Chem. C* 128, 8839 (2024).

Therefore, I cannot accept this manuscript for publication in its current form. The authors need to identify precisely how their new observations and models extend the existing literature of semi-ordered ternary oxides.

Minor comments

"Scanning electron microscopy (SEM) results reveal 2, 1 and 3 orientational growth of Nb-W-O nanorods on Nb-doped SrTiO₃ (001), (011) and (111) substrates, respectively." This statement is very unclear – what does "2 orientational growth" etc. mean? While I think I see what the authors are getting at, they should clarify that they mean families of nanowire orientation.

Figure 4 caption is way too long. All the captions are too long, but especially this one. Concisely describe your observations and move discussion to the main text.

"... defined by the choice of substrate, rather than that it is the result of an uncontrolled local phenomenon." Could be clarified.

Reviewer #4

(Remarks to the Author)

The authors have fabricated a unique three-dimensional structure composed of stacked two-dimensional amorphous layers in the Nb-W-O system. By taking advantage of the unidirectional periodicity of this structure, they performed direct imaging of the amorphous structure using state-of-the-art STEM techniques. This approach, combined with X-ray diffraction, EXAFS, and RSM analyses, enables a detailed structural characterization. A key strength of the study is that the electron beam was aligned parallel to the periodic axis, allowing unambiguous imaging of the amorphous structures. The experimental work is thorough, and the discovery of a 3D structure possessing 2D amorphous character is noteworthy. However, I have several concerns as listed below:

1. The authors should address the relationship between their findings and previous work, particularly Ref. 8 (Huang et al., *Nano Lett.* 12, 1081 (2012)), where a two-dimensional amorphous SiO₂ film was directly imaged using STEM and shown to reproduce the classic Zachariasen model. Although the current study focuses on a three-dimensional structure, the amorphous nature is still essentially two-dimensional. The authors should clearly articulate the scientific advancement beyond previous studies.
2. In the introduction, the authors refer to the longstanding debate on whether three-dimensional amorphous materials exhibit medium-range order (MRO). However, the present structure is fundamentally a two-dimensional amorphous material, and thus may not directly address this question. The authors are encouraged to comment on this point and clarify their position.
3. The manuscript claims that the amorphous layers exhibit only short-range order (SRO), without MRO. Is this conclusion fully justified? For example, the reduced pair distribution function $G(r)$ shown in Fig. 2f extends only up to 5 angstrom, making it difficult to conclusively determine the absence of MRO. Furthermore, the peak near 4.8 angstrom might correspond to in-plane amorphous features and could be indicative of MRO, especially since it exceeds typical interatomic distances.
4. In Figs. 3a and 3b, STEM images of samples grown on SrTiO₃ (001) and (011) substrates are shown, respectively. Fig. 3b appears more ordered than Fig. 3a. Could this reflect a difference in the degree of medium-range order? The authors should address this observation and provide their interpretation.
5. It is possible to extract atomic coordinates from the bright spots in the STEM images and calculate the reduced pair distribution function $G(r)$. I recommend comparing this with the $G(r)$ obtained from XRD measurements. This comparison would help disentangle the contributions from amorphous in-plane structure and periodicity along the stacking direction.
6. It remains unclear what conceptual or structural insights this study offers beyond the compatibility with Zachariasen's model. The authors are encouraged to more clearly state the broader implications of their findings for the structural science

of amorphous materials, particularly in the summary section.

Version 1:

Reviewer comments:

Reviewer #1

(Remarks to the Author)

This second version of the manuscript greatly improves the understanding of what the authors would demonstrate. After reading the new version, I believe the manuscript can be accepted for publication if the authors address the following two comments.

Comment 1:

According to one of my comment on the previous version, the authors have now defined a clear orientation notation with respect to both the measurement orientation and the crystallographic main directions. Unfortunately, they used the symbol “**” to label the [100], [010] and [001] directions. This is a pity because this symbol is used in crystallography to represent directions in the reciprocal space. Another choice has to be done.

Comment 2:

On page 2, the authors wrote that the material under study is amorphous in the x-y plane and this statement refers to the structural model reported Fig. 1 a-c. In fact, not any representation of structure in x-y plane is reported Fig. 1. The authors state that, due to the periodicity in the third direction, such an in plane structure cannot be represented. Nevertheless, in principle, a slice of the structure with a thickness of one atomic plane could evidence this in plane amorphous nature. This must be added.

Reviewer #2

(Remarks to the Author)

The reviewer has carefully checked the response to the comments on the manuscript entitled “Crystalline or Amorphous? A Matter of Perspective” by Xia et al.

Although most questions have been properly addressed, the reviewer is unsatisfied with the response to comment 4. In the response to comments 4, the author wrote “The amount of Nb-W-O nanorods is negligible compared to the substrate. Thus, it is impossible to detect the Nb-W-O signal in differential scanning calorimetry. Furthermore, collecting a large enough amount of nanorods from the substrate surface to enable differential scanning calorimetry seems extremely challenging”. In fact, this reply is not true for flash DSC. The desired sample amount for flash DSC is only a few micrograms, and the Nb-W-O signal can be detected by scraping a few pieces of nanorods from the substrate.

To determine the nature of this non-crystalline matter, the reviewer still recommends that the authors perform thermal measurements using flash DSC. However, this can be addressed in a future study. The manuscript has been significantly improved and it deserves the publication in Nature Communications.

Reviewer #3

(Remarks to the Author)

I am still not entirely convinced about the author's claims of <5% medium range order (MRO) or the novelty of this study. By eye in Figures 1 and 2 I think I see correlations well beyond 5 nm, and I agree with the other referee that the PDF should be extended to longer length scales. By taking these image and looping through all (masked) bright sites, the authors could easily compute an imaging PDF / reduced PDF.

However I feel the authors have done their due diligence to investigate most of questions of myself and the other referees. Therefore I am prepared to accept this manuscript for publication, if the authors can fulfill this request:

-Please upload all of the raw TEM images published in this study to a public repository (such as Zenodo). It's not acceptable to write "Data Availability - The data that support the findings of this study are available from the corresponding author upon reasonable request." anymore and Springer Nature policy requires upload of all data (and analysis codes), here is the policy:

<https://www.nature.com/nature-portfolio/editorial-policies/reporting-standards#availability-of-data>

If the authors won't measure the long range PDF themselves, they must make it possible for the research community to do so (and I will be checking your results after publication!).

Reviewer #4

(Remarks to the Author)

I appreciate the authors' revisions in response to my comments. In particular, the modifications regarding Comments 1, 2, 3, and 4 satisfactorily address my concerns.

However, I still have remaining concerns regarding my previous Comments 5 and 6.

1. With respect to my previous comment 5, thank you for the clarification regarding the difficulty of obtaining reliable SAED data for PDF analysis. However, I would like to emphasize that the main point of my earlier comment was not about deriving PDF from SAED, but rather about the possibility of extracting atomic coordinates directly from the STEM images and calculating the 2D reduced pair distribution function($G(r)$) from them. This approach does not rely on SAED and would make better use of the STEM data that the authors already possess. By comparing the STEM-derived $G(r)$ with the one obtained from XRD, the authors could provide a more quantitative basis for their discussion of medium-range order (MRO) and disentangle the respective contributions from in-plane amorphous structure and stacking periodicity. Such an analysis would significantly strengthen the manuscript and support the conclusions in a more rigorous way.

2. With respect to my previous comment 6, I appreciate the authors' efforts to revise the summary section. However, the response still does not fully address the essence of my concern. The main point was to clarify what conceptual or structural insights this study provides for the field of amorphous materials beyond demonstrating compatibility with Zachariassen's model. The revision largely restates the anisotropic order-disorder character of the nanorods, but it remains unclear how this observation advances broader understanding. In particular, the authors are encouraged to state more explicitly what the key factor is in distinguishing the boundary between amorphous and crystalline states, and what implications this may have for future studies of amorphous/crystalline systems.

Version 2:

Reviewer comments:

Reviewer #3

(Remarks to the Author)

While I still have some reservations, I believe the authors have addressed my comments to a degree satisfactory for publication. I expect this paper will contribute to significant discussion in the disordered materials community. I do understand the PDFs are just from the metal atoms, which is important to point out in the text. I believe these PDFs place the authors' claims on much firmer ground.

Reviewer #4

(Remarks to the Author)

The authors have addressed all my comments carefully and satisfactorily. The revised manuscript is significantly improved and is now suitable for publication in Nature communications. I have no further concerns.

Response to Reviewer #1

General comment: This paper deals with the elaboration and deep structural analysis of samples made of periodic stacking of niobium and tungsten oxide layers without in plane long range order. The authors describe this 1D ordering without in plane long range order as a new state of condensed matter, which they call “amorphous-crystalline”. Such ambition is of course very high and it corresponds to noteworthy results.

Some experimental aspects support the author’s assumption. Nevertheless, the manuscript is too hard to read, and figures are often very difficult to understand. The writing of all the paper has to be reconsidered in order to be more readable. One of the aspects that makes reading difficult is the continuous transition, throughout the text, between the orientation of the columns of atoms and the crystallographic orientation of the nanorods. In addition, often the authors refer their scientific comments to supplementary data or notes. This increases the difficulty for the readers. This aspect must be carefully reconsidered case by case.

A large part of the experimental results is strongly related to crystallography. As a result, a stronger full crystallographic notation and description must be used. Due to the fundamental influence of the crystallographic orientation of the substrate surface, all the crystallographic results (in the text and in the figures) have to be presented with respect to the crystallographic axis of these substrates instead of using the q_x , q_y and q_z vector components. These components are related to the measurement conditions, and therefore their meaning often changes, sometimes even in the same figure.

Response to general comment: We sincerely thank Reviewer 1 for the positive evaluation. We fully agree that the different coordinate systems, and corresponding descriptions, make it difficult for readers to understand. Following your insights, we unify the coordinate system in all the characterizations. Firstly, we introduced the main coordinate system $x^*-y^*-z^*$ from **Fig. 1**. We defined the two main coordinate systems in **Supplementary Note 1** for easier understanding.

The $x^*-y^*-z^*$ coordinate system is defined:

x^* is the [100] orientation of SrTiO_3 ,

y^* is the [010] orientation of SrTiO_3 ,

z^* is the [001] orientation of SrTiO_3 .

The q_x - q_y - q_z coordinate system is defined:

q_z is defined as a certain crystallographic direction of SrTiO_3 which is used as the

calibration peak for each High resolution XRD test (3D RSM test),

q_x and q_y are defined as the simplest crystallographic directions of SrTiO₃ which are in the perpendicular plane to the defined q_z and allow the input X-ray to detect the nanorods film.

Based on those, we revised **Fig. 2** and **Fig. 4** by adding the coordinate system $x^*-y^*-z^*$ for coherent description on different characterization results. Furthermore, we gave the definition on the 2 coordinate systems we used for descriptions in the captions of new **Fig. 1** and new **Fig. 4** for better understanding.

Based on those, we have revised **Fig. 1**, **Fig. 2**, and **Fig. 4** and added more information to the main text (line 80 to 96) for easier understanding, as shown in **Figure R1-2,5-6**. The modification of the supporting information related to this part can be seen in **Supplementary Note 1**.

Supplementary note 1

In this work, two coordinate systems are defined for coherent description though different characterization and simulation results.

The $x^*-y^*-z^*$ coordinate system is defined:

x^* is the [100] orientation of SrTiO₃,

y^* is the [010] orientation of SrTiO₃,

z^* is the [001] orientation of SrTiO₃.

This coordinate system is used in the description of SEM analysis, STEM analysis and 3D RSM simulation results.

The q_x - q_y - q_z coordinate system is defined:

q_z is defined as a certain crystallographic direction of SrTiO₃ which is used as the calibration peak for each High resolution XRD test (3D RSM test),

q_x and q_y are defined as the simplest crystallographic directions of SrTiO₃ which are in the perpendicular plane to the defined q_z and allow the input X-ray to detect the nanorods film.

The q_x - q_y - q_z coordinate system is used in the description of 3D RSM test and the matched simulation results. Therefore, q_x - q_y - q_z coordinate systems are different in the five 3D RSM tests:

1. For the 3D RSM test of Nb-W-O nanorods on SrTiO₃ (011) and its matched simulation, q_z is defined as [011] orientation of SrTiO₃. Then, q_x and q_y are aligned with [100], [0-11] orientation of SrTiO₃, respectively.
2. For the 3D RSM test of Nb-W-O nanorods on SrTiO₃ (001) and its matched simulation, q_z is defined as [001] orientation of SrTiO₃. Then, q_x and q_y are aligned with [100], [010] orientation of SrTiO₃, respectively.
3. For the 3D RSM test of Nb-W-O nanorods on SrTiO₃ (111) at testing direction 1 (SrTiO₃ (111) peak as the calibration peak of the 3D RSM test) and its matched simulation, q_z is defined as [111] orientation of SrTiO₃. Then, q_x and q_y are aligned with [1-10], [11-2] orientation of SrTiO₃, respectively.
4. For the 3D RSM test of Nb-W-O nanorods on SrTiO₃ (111) at testing direction 2 (SrTiO₃ (011) peak as the calibration peak of the 3D RSM test) and its matched simulation, q_z is defined as [011] orientation of SrTiO₃. Then, q_x and q_y are aligned with [100], [0-11] orientation of SrTiO₃, respectively.
5. For the 3D RSM test of Nb-W-O nanorods on SrTiO₃ (111) at testing direction 3 (SrTiO₃ (001) peak as the calibration peak of the 3D RSM test) and its matched simulation, q_z is defined as [001] orientation of SrTiO₃. Then, q_x and q_y are aligned with [1-10], [110] orientation of SrTiO₃, respectively.

Figure R1 (new Fig.1) Schematics and SEM images of different type Nb-W-O nanorod films, a 3D Structure of Nb-W-O nanorods (z axis is the growth direction), **b** Transverse section of Nb-W-O nanorods, **c** Longitudinal section of Nb-W-O nanorods, **d** Schematic of the Nb-W-O nanorods on SrTiO₃ (001) substrate, **e-f** Top and side view SEM images of the Nb-W-O nanorods on SrTiO₃ (001) substrate, **g** Schematic of the Nb-W-O nanorods on SrTiO₃ (011) substrate, **h-i** Top and side view SEM images of the Nb-W-O nanorods on SrTiO₃ (011) substrate, **j** Schematic of the Nb-W-O nanorods on SrTiO₃ (111) substrate, **k-l** Top and side view SEM images of the Nb-W-O nanorods on SrTiO₃ (111) substrate. (In this work, x*, y*, z* are defined as the [100], [010], [001] direction of SrTiO₃, details are described in **Supplementary Note 1.**)

Comment 1: Page 4: according to page 4, the image reported fig2a, is a plane view and in this case (SrTiO₃ (011) substrate) the Nb-W-O nanorods have grown in the plane, along the [100] direction (see Fig. 1g). It seems to mean that the atom columns are along the [011] direction. This must be described in an easier understanding way.

Response to comment 1: We agree that “plane view” and “side view” are not very clear to the reader. Thus, we modified **Fig.2**, changed the terms for each STEM image and added schematic for the FIB cut direction in each STEM image. We also added substrate information in the new **Fig. 2** for better reading. The FIB cutting direction is also explained as a **Supplementary Note 4**.

Fig.2a is captured at “transverse section of Nb-W-O nanorods on SrTiO₃ (011) substrate” and “[100] projection of the SrTiO₃ (011) substrate”. **Fig.2c** is captured at “longitudinal section of Nb-W-O nanorods on SrTiO₃ (011) substrate” and “[0-11] projection of the SrTiO₃ (011) substrate”. **Fig.2e** is captured at “[100] projection of the SrTiO₃ (001) substrate”. Thus, the atom columns of the Nb-W-O nanorods on SrTiO₃ (011) substrate are along the [100] direction of SrTiO₃ and the Nb-W-O nanorods on SrTiO₃ (011) and SrTiO₃ (001) substrate have grown in the substrate plane.

The modification of the manuscript related to this part can be seen in line 97 to 112 and **Fig.2**. The modification of the support information related to this part can be seen in **Supplementary Note 4**.

Figure R2 (New Fig. 2) STEM analysis of atomic ordering in Nb-W-O nanorod films, a STEM image of the transverse section of Nb-W-O nanorods on SrTiO₃ (011) substrate, **b** Fast Fourier transform image of **a**, **c** STEM image of the longitudinal section of Nb-W-O nanorods on SrTiO₃ (011) substrate, **d** Fast Fourier transform image of **c**, **e** STEM image of Nb-W-O nanorods on SrTiO₃ (001) substrate, **f** Zoomed in STEM image of **e** at film-substrate interface. (The red line shows the boundary of the two directions in Nb-W-O. The blue box represents the first epitaxial Nb-W-O layer. The yellow circle indicates the step of the substrate.), **g** PDF of Nb-W-O nanorods on SrTiO₃ (001), SrTiO₃ (011), and SrTiO₃ (111) substrates, **h-i** EXAFS data of Nb (**h**) and W (**i**) of Nb-W-O nanorods on SrTiO₃ (001) substrate.

Supplementary Note 4

Sample in **Fig.2a** was cut along the SrTiO₃ (100) plane using FIB to expose this transverse section of the nanorods (shown in **Figure S6a**). Therefore, in **Fig. 2a**, the transverse section of Nb-W-O nanorods on the SrTiO₃ (011) substrate lies parallel to the SrTiO₃ (100) plane.

Sample in **Fig. 2c** was cut along the SrTiO₃ (0-11) plane using FIB to expose this longitudinal section of the nanorods (shown in **Figure S6b**). Therefore, in **Fig. 2c**, the

longitudinal section of Nb-W-O nanorods on the SrTiO₃ (011) substrate lies parallel to the SrTiO₃ (0-11) plane.

Sample in **Fig. 2e** was cut along the SrTiO₃ (100) plane using FIB to expose both the transverse and longitudinal sections of the nanorods (shown in **Figure S6c**). Therefore, in **Fig. 2e**, the transverse and longitudinal sections of Nb-W-O nanorods on the SrTiO₃ (001) substrate are both lies parallel to the (010) and (100) plane of the SrTiO₃ (001) substrate due to the orthogonal in plane growth of thw Nb-W-O nanorods on the SrTiO₃ (001) substrate.

Figure R3 (Figure S6) Focused ion beam (FIB) cut schematic for **a** sample 1 present in **Fig.2 a**, **b** sample 2 present in **Fig.2 c**, **c** sample 3 present in **Fig.2 e**.

Comment 2: On Fig. 2, the images reported in a, b, c and d are corresponding to observation of the columns that have grown on the SrTiO₃(011) substrate. Please, separate this case from that one reported in fig. 2e, that is related to the interface with a SrTiO₃(001) substrate. In the current state, Fig. 2 is really hard to understand. Moreover, the main crystallographic directions must be reported on the images.

Response to comment 2: We agree that **Fig.2** is not clear enough. We revised Fig.2 shown in Figure R2. In the **new Fig. 2**, we added substrate information on each STEM image. Also, we added the crystallographic orientation of the SrTiO₃ to explain the projection of the STEM images for easier understanding.

Comment 3: The contrast observed on Fig. 2c must be discussed. How can a 1D atomic ordering results on such contrast in the image in the TEM? Such image, that is fundamentally an interferences image, comes from a volume of matter. Please give some explanations of the observation in direct space (high resolution TEM image) of a periodic arrangement in transmission mode in such a case as only a 1D periodicity. Moreover, the Fourier transform (Fig. 2d) of this image seems to evidence the presence of regularly spaced points along the straight diffuse lines.

Response to comment 3: All TEM images are HAADF-STEM images, which is an incoherent imaging technique without interference. It directly reveals atomic columns and enables atom-by-atom analysis. FFT pattern is likely due to slight instrument instability.

Comment 4: Page 10 and 11, Fig.4: 3D RSM are recorded on the sample. According to the size of the x-ray beam, each RSM is due to a large number of columns. How the authors have separated the influence of both the intra-nanorods disorder from the rotational disorder coming to the fact that a large number of nanorods are simultaneously probed?

Response to comment 4: We agree with reviewer 1 that a large number of nanorods are simultaneously probed. We roughly calculated the number of the nanorods detected to be 10^4 (**Figure S71**) by the diameter of X-ray beam (2.92 mm) and diameter of the nanorod (30 nm).

The observed disorder of the nanorods from RSM has two possible origins:

1. the intra-nanorods disorder;
2. the intrinsic disorder in the nanorod.

First, we observe and confirm the disorder along the transverse section of the nanorods in the STEM images in **Fig. 2**.

Second, the nanorods in the detection range can be observed in the SEM images (**Figure R4**). All the nanorods are grown in a fixed direction:

Along SrTiO₃ [100] and [010] orientations on SrTiO₃ (001) substrate,

along SrTiO₃ [100] orientation on SrTiO₃ (011) substrate,

along SrTiO₃ [100], [010] and [001] orientations on SrTiO₃ (111) substrate.

For samples grown on SrTiO₃ (001) and SrTiO₃ (011) substrates, the intra-nanorod disorder due to rotational disorder is not possible, since the nanorods are grown parallel to the substrate.

Rotational disorder is only possible for samples grown on SrTiO₃ (111) substrates, although the nanorods in this sample are fixed in three directions (SrTiO₃ [100], [010] and [001] orientations) and no rotation is observed.

Figure R4 a-c Low magnification SEM image of Nb-W-O nanorods on SrTiO₃ (001), (011) and (111) substrates.

Comment 5: Fig. 4 is very hard to read. As written above, the first improvement is to replace the q_x , q_y and q_z vectors notation by the true crystallographic directions in the reciprocal space. The relationships between the Q-vector components and the crystallographic directions depend on the growth direction, that itself differs when the three types of substrate cutting are considered. In the present form, the notations reported in this figure are too confusing.

Response to comment 5: We agree that the simulation and experimental analysis of the reciprocal space structural information investigated by 3D RSM and RHEED analyses of the Nb-W-O nanorods is an important section of our manuscript, and we need to expand this section in detail in the main text. Thus, we revised this section into two separate Figures in the revised manuscript to emphasize the importance of this section and to improve its readability.

In the new **Fig.4 (Figure R5)** we added the schematic of the 3D RSM analysis and the 3D RSM simulation in 3D RSM analysis area. We transferred the 3D RSM analysis results to the additional figure. The schematic of the 3D RSM analysis will help the reader to understand the relationships between x^* , y^* , z^* and q_x , q_y , q_z coordinates.

The $x^*-y^*-z^*$ coordinate system is defined:

x^* is the [100] orientation of SrTiO₃,

y^* is the [010] orientation of SrTiO₃,

z^* is the [001] orientation of SrTiO₃.

The $q_x-q_y-q_z$ coordinate system is defined:

q_z is defined as a certain crystallographic direction of SrTiO₃ which is used as the calibration peak for each High resolution XRD test (3D RSM test),

q_x and q_y are defined as the simplest crystallographic directions of SrTiO₃ which are in the perpendicular plane to the defined q_z and allow the input X-ray to detect the nanorods film. The detailed explanation of the defined coordinate systems is in **Supplementary Note 1**.

In the new **Fig.5 (Figure R6)** we added the q_x-q_z projection and q_x-q_y slice at different q_z of the 3D RSM simulation results for direct comparison to the 3D RSM experimental results.

Figure R5 (new Fig.4) 3D Reciprocal space mapping (RSM) simulation for Nb-W-O nanorod films, a 3D RSM simulation for the stack of single amorphous layers, **b** q_x - q_z projection of **a**, **c** q_y - q_z projection of **a**, **d-f** Schematic of 3D RSM analysis of Nb-W-O nanorods on SrTiO₃ (011), (001) and (111) substrates respectively*, **g-i** 3D RSM simulation for Nb-W-O nanorods on SrTiO₃ (011), (001) and (111) substrates respectively, **j-l** Results of **g-i** in 3D RSM analysis area respectively. (*In this work, q_x , q_y , q_z are defined by the calibration peak of SrTiO₃ which is applied for each 3D RSM analysis, as described in **Supplementary Note 1.**)

Figure R6 (new Fig.5) 3D RSM analysis of structural ordering in Nb-W-O nanorod films, a-c q_x - q_z projection and q_x - q_y slice at different q_z of the 3D RSM simulation results in the 3D RSM analysis area for Nb-W-O nanorods on SrTiO₃ (011), (001) and (111) substrates (**Fig.4 j-l**) respectively, **d-f** q_x - q_z projection and q_x - q_y slice at different q_z of the experimental 3D RSM of Nb-W-O nanorods on SrTiO₃ (011), (001) and (111) substrates respectively.

We still retain “ q_x , q_y , q_z ” in the figure based on the standard convention for RSM Figures as shown for example in different references (*Nature Communications*, 2023, 14, 8496; *Journal of Applied Crystallography*, 2014, 47(2), 762–769). When we report a RSM image, we usually observe the crystallographic information from the substrate as well as the film, and it is not clear to use only one of their crystallographic orientations as the axis. Thus, we added the explanation of q_x , q_y , q_z coordinates for each 3D RSM simulation and experimental analysis for easier understanding.

The modification of the manuscript related to this part can be seen in line 241 to 334 and Fig.4-5.

Response to Reviewer 2

General comment: The manuscript entitled “Crystalline or Amorphous? A Matter of Perspective” by Xia et al. demonstrates an interesting inorganic matter which is amorphous in two dimensions but crystalline in the third. The crystalline (including quasicrystal) and amorphous structures are two states for solid matters. They are commonly believed to be mutually exclusive that a state of matter cannot be both crystalline and amorphous at the same time. Usually, a partially crystalline or amorphous matter is just a composite of purely crystalline and amorphous parts. Hence, the matter demonstrated in this manuscript presents experimentally a novel state which is only ordered in certain dimensions.

In fact, such a state of matter has been theoretically proposed as paracrystalline. The term paracrystal refers to a state of matter with less than three-dimensional order, characteristic of a true crystal (Ref: <https://www.merriam-webster.com/dictionary/paracrystal>). Paracrystals are often used in materials science to understand the transition between crystalline and amorphous states. Although the paracrystalline matter has been demonstrated microscopically in many papers (Science 335, 950, (2012); Nature 599, 605 (2021); JAP 90, 4438 (2001)), a direct observation of such matter in bulk is still missing. In this manuscript, we can see the micrometer-scale formation of paracrystalline matter, and the structural characterizations provide compelling evidence in a degree of intermediate order between crystalline and amorphous materials. This manuscript contributes significantly to the existing knowledge on the state of solid matter.

Response to the general comment: We agree with reviewer 2 that the structure of Nb-W-O nanorods falls under the definition of paracrystal. The definition of paracrystal is “a solid body with less than three-dimensional order characteristic of a true crystal”. (Ref: <https://www.merriam-webster.com/dictionary/paracrystal>) Basically, everything between pure amorphous phase and crystal phase fits this definition. Thus, it does not conflict with our structural description (amorphous in two dimensions but crystalline in the third). Furthermore, a 2D continuous random network amorphous structure also allows small proportion of medium-range order inside of its structure, as confirmed by Voyles et al. (*Journal of Applied Physics*, 2001, 90 (9), 4437.): “...is a signature of structures with little or no medium-range order, such as a continuous random network”.

Therefore, we believe that our study provides new insight to distinguish the boundary between amorphous and crystalline cases, for which the direction matters. This is beyond current understanding in the debate how medium range disorder determines the boundary between amorphous and crystalline materials.

Additional detailed analysis of the order parameters as well as the medium-range order has been described in the new supplementary notes 8 and 9, and Figures S38-S42.

Supplementary note 8

Order parameter calculation for 2D amorphous plane of the Nb-W-O nanorods

The scalar order parameter described by Tsvetkov quantifies the ordering of nematic molecules behaving as rod-like ellipsoids or rigid rods (*Molecular Crystals and Liquid Crystals*, 2002, 381(1), 1–19.):

$$S = \frac{1}{2}(3\cos^2\theta - 1) \quad \text{SN8.1}$$

where θ is the angle between the long molecular axis and the optic axis, shown in **Figure R7**.

Figure R7 (Figure S38) Schematic of angle θ between the long molecular axis and the optic axis.

For a uniaxial rod-like ellipsoids or rigid rods, the order parameter of the liquid crystal system can be described by a function with only one variable (θ) which presents the angle between the long molecular axis and the optic axis. Because of the inversion symmetry of the rod-like ellipsoids or rigid rods, θ and $\pi - \theta$ are indistinguishable. Thus, second order Legendre polynomial can be used to describe the order parameter of rod-like ellipsoids or rigid rods.

Based on the above definition, the order parameter of different rings in Nb-W-O 2D amorphous plane can also be described.

For liquid crystal such as rod-like ellipsoids or rigid rods, a certain optic axis can be used as the most ordered direction to easily calculate the order parameter by using the angle between the long molecular axis and the optic axis. However, in the 2D amorphous plane of the Nb-W-O nanorods, multiple ring like structures are the basic unit, which does not have a certain axis that can be used as the most ordered direction. Thus, we need to assume any direction can be the most ordered direction. Thus, we define the angle α as an angle between the ordered direction and the normal direction. And we define the angle θ_i as the angle between the direction of every ring and the normal direction ($i \in [1, m]$ & $i \in \mathbb{Z}, m$ is the total number of the rings), as shown in **Figure R8**.

Figure R8 (Figure S39) Schematic of the order parameter definition for the (3,0) rings in the 2D amorphous plane of the Nb-W-O nanorods.

Order parameter for (3,0) rings

As shown in **Figure R9**, (3,0) ring is a regular triangle and has $c4$ symmetry in the $\text{Nb}_{18}\text{W}_{16}\text{O}_{93}$ crystal. Thus, in principle, we need to use the twelfth order Legendre polynomial to describe the order parameter, which is a complex equation. In this case, we can process the obtained $\theta_{(3,0)i}$ by the following equation:

$$\theta_{(3,0)i}^1 = \theta_{(3,0)i} - \frac{n\pi}{6} \left(n \in \mathbb{Z}, 0 \leq \theta_{(3,0)i}^1 \leq \frac{\pi}{6} \right) \quad \text{SN8.2}$$

After processing the obtained $\theta_{(3,0)i}$, we can use the first order Legendre polynomial to write the function between order parameter and α :

$$S_{(3,0)}(\alpha) = \left| \frac{\sum_{i=1}^m \cos(12(\theta_{(3,0)i}^1 - \alpha))}{m} \right| \quad \alpha \in \left[0, \frac{\pi}{6}\right) \quad \text{SN8.3}$$

And we can describe the order parameter:

$$S_{(3,0)} = \max(S_{(3,0)}(\alpha)) = \max\left(\left|\frac{\sum_{m=1}^m \cos(12(\theta_{(3,0)m}^1 - \alpha))}{m}\right|\right) \quad \alpha \in \left[0, \frac{\pi}{6}\right) \quad \text{SN8.4}$$

To verify the above equation, we calculated the maximum and minimum of the equation SNx.4.

When the distribution of $\theta_{(3,0)m}^1$ are a concentrated distribution into one angle θ , $S_{(3,0)}(\alpha)$ has its maximum when $\alpha = \theta$:

$$S_{(3,0)}(\alpha) = \left| \frac{\sum_{m=1}^m \cos(12(\theta - \alpha))}{m} \right| = \left| \frac{\sum_{m=1}^m 1}{m} \right| = 1 \quad \text{SN8.5}$$

When the distribution of $\theta_{(3,0)m}^1$ are an even distribution in the range of $\theta_{(3,0)m}^1$, $S_{(3,0)}(\alpha)$ has its maximum and α can be any value in its' range:

$$\begin{aligned} S_{(3,0)}(\alpha) &= \left| \frac{\sum_{m=1}^m \cos(12(\theta - \alpha))}{m} \right| = \left| \frac{\int_{0-\alpha}^{\frac{\pi}{6}-\alpha} \cos(12\theta_{(3,0)m}^1) d\theta_{(3,0)m}^1}{m} \right| \\ &= \left| \frac{\frac{1}{12}(\sin(2\pi - 12\alpha) + \sin(12\alpha))}{m} \right| = 0 \end{aligned} \quad \text{SN8.6}$$

Thus, when the distribution of $\theta_{(3,0)m}^1$ are a concentrated distribution, $S_{(3,0)}$ is 1.

When the distribution of $\theta_{(3,0)m}^1$ are an even distribution, $S_{(3,0)}$ is 0. The above results show that the equation of $S_{(3,0)}$ can describe the order parameter of (3,0) rings nicely.

Figure R9 (Figure S40) Schematic of the Nb and W atom placement in $\text{Nb}_{18}\text{W}_{16}\text{O}_{93}$ crystal.

Order parameter for (4,0) and (5,0/1) rings

Order parameters for the other rings are described similar to the (3,0) rings.

As shown in **Figure R9**, (4,0) ring is a square and has $c8$ symmetry in the $\text{Nb}_{18}\text{W}_{16}\text{O}_{93}$ crystal. Thus, in principle, we need to use the eighth order Legendre polynomial to describe the order parameter, which is a complex equation. In this case, we can process the obtained $\theta_{(4,0)i}$ by the following equation:

$$\theta_{(4,0)i}^1 = \theta_{(4,0)i} - \frac{n\pi}{4} \quad \left(n \in \mathbb{Z}, 0 \leq \theta_{(4,0)i}^1 \leq \frac{\pi}{4} \right) \quad \text{SN8.7}$$

After processing the obtained $\theta_{(4,0)i}$, we can use the first order Legendre polynomial to write the function between order parameter and α :

$$S_{(4,0)}(\alpha) = \left| \frac{\sum_{i=1}^m \cos(8(\theta_{(4,0)i}^1 - \alpha))}{m} \right| \quad \alpha \in \left[0, \frac{\pi}{4} \right) \quad \text{SN8.8}$$

And we can describe the order parameter:

$$S_{(4,0)} = \max(S_{(4,0)}(\alpha)) = \max \left(\left| \frac{\sum_{i=1}^m \cos(8(\theta_{(4,0)i}^1 - \alpha))}{m} \right| \right) \quad \alpha \in \left[0, \frac{\pi}{4} \right) \quad \text{SN8.9}$$

As shown in **Figure R9**, the direction of (5,0) rings and (5,1) rings are the same. Thus, we consider using one order parameter to describe the 5 side rings ((5,0/1) rings). As

shown in **Figure R9**, (5,0/1) ring is a regular pentagon and has $c4$ symmetry in the $\text{Nb}_{18}\text{W}_{16}\text{O}_{93}$ crystal. Thus, in principle, we need to use the twentieth order Legendre polynomial to describe the order parameter, which is a complex equation. In this case, we can process the obtained $\theta_{(5,0/1)i}$ by the following equation:

$$\theta_{(5,0/1)i}^1 = \theta_{(5,0/1)i} - \frac{n\pi}{10} \left(n \in \mathbb{Z}, 0 \leq \theta_{(5,0/1)i}^1 \leq \frac{\pi}{10} \right) \quad \text{SN8.10}$$

After processing the obtained $\theta_{(5,0/1)i}$, we can use the first order Legendre polynomial to write the function between order parameter and α :

$$S_{(5,0/1)}(\alpha) = \left| \frac{\sum_{i=1}^m \cos(20(\theta_{(5,0/1)i}^1 - \alpha))}{m} \right| \alpha \in \left[0, \frac{\pi}{10} \right) \quad \text{SN8.11}$$

And we can describe the order parameter:

$$S_{(5,0/1)} = \max(S_{(5,0/1)}(\alpha)) = \max\left(\left|\frac{\sum_{i=1}^m \cos(20(\theta_{(5,0/1)i}^1 - \alpha))}{m}\right|\right) \alpha \in \left[0, \frac{\pi}{10} \right) \quad \text{SN8.12}$$

Results for order parameter calculation

In **Fig.3**, we collected the direction of (3,0), (4,0) and (5,0/1) rings (angle θ in **Figure R2**). Thus, we can use them for the calculation of the order parameter.

Figure R10 a-c (Figure S41) Order parameter curve with altered α for (3,0), (4,0) and (5,0/1) rings respectively in 2D amorphous plane of Nb-W-O nanorods on SrTiO_3 (001) and (011) substrate.

As shown in **Figure R10**, each curve has two maximum points, the angle α at S_{max} means the actual ordered direction for its matched rings. Thus, the order parameter of (3,0), (4,0) and (5,0/1) rings in 2D amorphous plane of Nb-W-O nanorods on SrTiO_3 (001) and (011) substrates (**Fig.3 a-b**) are listed in **Table 1**.

Table 1 Order parameter for (3,0), (4,0) and (5,0/1) rings in 2D amorphous plane of Nb-W-O nanorods on SrTiO₃ (001) and (011) substrates

	$S_{(3,0)}$	$S_{(4,0)}$	$S_{(5,0/1)}$
Nb-W-O on SrTiO ₃ (001)	0.031	0.139	0.077
Nb-W-O on SrTiO ₃ (011)	0.068	0.040	0.072

As shown in **Table 1**, the order parameters of every different ring in 2D amorphous plane of Nb-W-O nanorods on both SrTiO₃ (001) and (011) substrates are quite small, which shows high randomness of the amorphous plane of Nb-W-O nanorods on SrTiO₃ (001) and (011) substrates. Higher order parameters are revealed in (3,0) ring on SrTiO₃ (011) substrate and (4,0) ring on SrTiO₃ (001) substrate and the order parameter for (5,0/1) ring is similar in both cases. Due to the inconsistency of the comparison of the order parameters in both cases and the low order parameters for every ring structure in the 2D amorphous plane of Nb-W-O nanorods, the randomness of 2D amorphous plane of Nb-W-O nanorods can be similar and high for both cases.

Supplementary note 9

medium-range order area analysis

Figure R11 a-b (Figure S42) medium-range order analysis for **Fig.3 a-b** respectively.

Based on the examples shown by reviewer 2, we find the medium-range order area in **Fig.3 a-b**. As shown in **Figure R11a**, in the STEM image of the 2D amorphous plane of Nb-W-O nanorods on SrTiO₃ (001), the medium-range order area proportion of the total STEM image is 1.1%. As shown in **Figure R11b**, in the STEM image of the 2D

amorphous plane of Nb-W-O nanorods on SrTiO₃ (011), the medium-range order area proportion of the total STEM image is 3.9%. Comparing the medium-range order area proportion, we can conclude that the randomness of the 2D amorphous plane on SrTiO₃ (011) is less than the one on SrTiO₃ (001). However, both medium-range order area proportions are small and considering the small order parameters, we can conclude that the 2D plane of Nb-W-O nanorods is amorphous with continuous random network (continuous random network) structure (*Nano Letter*, 2012, 12, 1081-1086; *Nature*, 2020, 577, 199–203). This randomness of the plane would be similar when we use the same growth conditions even with differently oriented SrTiO₃ substrates.

Three example cases from literature are shown in Figures **R12-R14** and indicate the common existence of medium-range order in 2D continuous random network amorphous materials. Also the appearance of a crystalline pattern in the FFT of the medium-range order area is a common phenomenon.

[Figure Redacted]

Figure R12 medium-range order in 2D continuous random network amorphous plane. (case 1, *Nano Letter*, 2012, 12, 1081-1086)

[Figure Redacted]

Figure R13 medium-range order in 2D continuous random network amorphous plane.
(case 2, *Nature*, 2020, 577, 199–203)

[Figure Redacted]

Figure R14 a certain medium-range order area in 2D continuous random network amorphous plane, **b** FFT of the medium-range order area 2D continuous random network amorphous plane. (case 3, *Nano Letter*, 2012, 12, 1081-1086)

Reviewer has carefully examined the manuscript and supports its publication in Nature Communications. However, it will be shown in the following comments why this matter should be considered as paracrystalline rather than amorphous in one direction.

Comment 1:

In Fig. 1, Nb-W-O nanorod films were deposited on crystalline substrates of different orientations. It is a smart approach to demonstrate the difference in structures between different orientations of a matter by showing the various growth morphologies. However, why do these rods all have a diameter around 50 nanometers? If it is truly amorphous in the x-y plane, the diameters of these rods should be infinite or distributed randomly.

Response to comment 1: The amorphous plane is the transverse section of the Nb-W-O nanorods. It is not always the x-y plane. In **Figure R15**, low magnification SEM images of Nb-W-O nanorods on SrTiO₃ (001), (011) and (111) substrates were used for diameter analysis. As shown in **Figure R15 a-c**, numerous nanorods are merged and nanorods with different widths are visible in the images. As shown in **Figure R15d**, the diameter widths of Nb-W-O nanorods on different substrates are all distributed in a large range (10-100 nm, 30-100 nm and 10-60 nm respectively).

Figure R15 a-c SEM images of Nb-W-O nanorods on SrTiO₃ (001), (011) and (111) substrates respectively, **d** diameter distribution of Nb-W-O nanorods on SrTiO₃ (001), (011) and (111) substrates.

Comment 2: What would happen if the films were deposited on an amorphous substrate?

Response to comment 2: The d-spacing of our Nb-W-O nanorods matches with the d-spacing of [001], [010] and [100] directions of SrTiO₃ substrate crystal, which forces the nanorods to grow along these directions (crystal axis are along these directions). Thus, if Nb-W-O are grown on amorphous substrates, we expect no nanorods structure will appear on the substrate for the growth conditions used.

Comment 3: In Fig. 2a and b, the author tries to prove the amorphous structure of the x-y plane by showing its STEM image and the corresponding FFT pattern. However, when the review carefully checks them, some tiny crystalline features can be observed as shown in the following Figures 1R, 2R and 3R. Hence, the x-y plane cannot be understood as fully amorphous but paracrystalline phase.

Fig. 1R. (a) STEM image of Fig. 2a in manuscript and (b) its corresponding halo FFT pattern. If we tuned the intensity of the FFT pattern, some diffraction spots corresponding to crystalline phase can be observed.

Fig. 2R. If we zoom in Figure 2a, a lot of tiny crystals can be observed.

Fig. 3R. Another region where the tiny crystals can be observed. Such regions are distributed over the whole image.

Response to comment 3: As we discussed above, everything between pure amorphous phase and crystal phase fits under this definition of a paracrystal and we agree with reviewer 2. However, we insist that the Nb-W-O nanorods exhibit a structure with 2D continuous random network amorphous plane together with a crystal axis in the third dimension.

As we discussed in **Supplementary Note 9**, we fully agree with reviewer 2 that medium-range order exists in the amorphous plane of Nb-W-O nanorods. However, it is very common for amorphous materials to possess certain medium-range order (*Nano Letter*, 2012, 12, 1081-1086; *Nature*, 2020, 577, 199–203). This is a hot and long-lasting topic in debating the distinct boundary between amorphous and crystalline materials. However, we provide new insight to distinguish the boundary between amorphous and crystalline cases, for which the direction matters.

Comment 4: The most important feature of amorphous matter differing from crystalline matter is the glass transition. Can the glass transition be observed in this matter by differential scanning calorimetry? If not, then this matter is paracrystalline rather than amorphous.

Response to Comment 4: We fully agree that differential scanning calorimetry would be interesting to understand the Nb-W-O nanorods. However, Nb-W-O nanorods (thickness: 50-200 nm) are grown on SrTiO₃ single crystal substrates (thickness: 0.5 mm). The amount of Nb-W-O nanorods is negligible compared to the substrate. Thus, it is impossible to detect the Nb-W-O signal in differential scanning calorimetry. Furthermore, collecting a large enough amount of nanorods from the substrate surface to enable differential scanning calorimetry seems extremely challenging.

Comment 5: By the way, some typos can be seen in the manuscript. “F PDF of Nb-W-O nanorods”.

Response to comment 5: Thanks for the critical reading. We revised the caption in **Figure R2 (new Fig. 2)**.

Comment 6: In summary, the state of matter in this manuscript is still unclear. The author needs to verify which kind of structure it is, amorphous or paracrystalline?

Response to comment 6: As discussed above, we believe that the structure of Nb-W-O nanorods falls under the definition of paracrystal. The state of the Nb-W-O matter shows the stacking of the 2D amorphous layer. In the matter along the amorphous plane, limited medium-range order is observed with continuous random network, while in the vertical direction, the layers are stacked at the specific layer spacing of 3.91 Å. We fully agree that the finding of our structure fits within the paracrystalline definition. However, we believe that the contribution of our case is to provide new insight to distinguish the boundary between amorphous and crystalline cases, for which the direction matters. This is beyond current understanding in the debate how medium range disorder determines the boundary between amorphous and crystalline materials.

Response to Reviewer #4

General Comment: This paper is interesting, though perhaps not as generally applicable as the authors imply in their abstract. The manuscript is well-written, and the characterization work seems solid. However, as I describe below, the authors need to update their knowledge of oxide structures beyond the year 1932 when Zachariasen published his seminal work on glass structures.

Comment 1:

My primary issue with this paper is that it's not as original as the author's claims imply. While the atomic resolution characterization in the current work is excellent, I have seen very similar structures (seemingly ordered along one axis) in several previous manuscripts: (none of which are cited by the authors)

-Miran Mozetič, Uros Cvelbar, Mahendra K. Sunkara, and Sreeram Vaddiraju, A Method for the Rapid Synthesis of Large Quantities of Metal Oxide Nanowires at Low Temperatures, *Adv. Mater.* 17, 2138 (2005).

-Subhra Jana and Robert M. Rioux, Seeded growth induced amorphous to crystalline transformation of niobium oxide nanostructures, *Nanoscale*, 4, 1782 (2012).

-Sophia B. Betzler, Ai Leen Koh, Bettina V. Lotsch, Robert Sinclair, Christina Scheu, Atomic Resolution Observation of the Oxidation of Niobium Oxide Nanowires: Implications for Renewable Energy Applications, *ACS Appl. Nano Mater.* 3, 9285 (2020).

Response to comment 1: Nb-based oxides and W-based oxide usually form a rod-like structure, as shown in **Figure R16 a-b** (*Nature*, 2018, 559, 556-563). In the references mentioned by reviewer 4 above (*Advanced Materials*, 2005, 17, 2138; *Nanoscale*, 2012, 4, 1782; *ACS Applied Nano Materials*, 2020, 3, 9285) TEM images show line like structure in all three works and SAED patterns indicate a nanocrystalline phase for all of them, as shown in **Figure R16 c-e**. In fact, the original crystalline material $\text{Nb}_{18}\text{W}_{16}\text{O}_{93}$ shows a line like structure in the [010] direction of $\text{Nb}_{18}\text{W}_{16}\text{O}_{93}$ crystal, as shown in the schematic of atomic ordering of $\text{Nb}_{18}\text{W}_{16}\text{O}_{93}$ crystal (**Figure R16 g**). Thus, the STEM image of the longitudinal section of the nanorods is only evidence for a fixed d-spacing along the longitudinal direction but no proof of the structural similarity. Moreover, the SAED patterns of the references above show a nanocrystalline phase for all three nanowire works, which is different from our findings.

[Figure Redacted]

Figure R16 a-b SEM image of $\text{Nb}_{18}\text{W}_{16}\text{O}_{93}$ crystal (*Nature*, 2018, 559, 556-563), **c-e** TEM and SEAD results from the reference mentioned by reviewer 4 (*Advanced Materials*, 2005, 17, 2138; *Nanoscale*, 2012, 4, 1782; *ACS Applied Nano Materials*, 2020, 3, 9285), **f-g** Structural schematic of $\text{Nb}_{18}\text{W}_{16}\text{O}_{93}$ crystal.

Comment 2:

Given that the current paper primarily focuses on structure, I expect to see a much deeper dive into other imaging studies on similar nanowires / materials.

Response to comment 2: We added the comparison of the atomic structure of Nb-W-O nanorods with traditional Nb-based and W-based nanowire/nanorod structures. The modification of the manuscript related to this part can be seen in line 110 to 112.

Comment 3: In fact, I believe the most similar crystallographic system to the current study isn't mentioned by the authors at all: 2D quasicrystals / (semi)rational approximates in ternary oxides. Figures 3a, b, and especially Figure 3c look like 2D decagonal quasicrystals to me. I suspect there may be additional disorders introduced by substrate and growth effects which may add some randomness to the quasicrystal lattice. For example, in Figure 2a, the (pseudo)coherent domain length appears to be only ~4 nm. Either way, the authors certainly need to explain exactly how their proposed structural models are similar to and distinct from 2D stacked quasicrystals. Here are a few references to get them started (there are many possible references here)

-https://link.springer.com/content/pdf/10.1007/978-3-642-58434-3_3 (textbook chapter)

-Stefan Förster et al., Quasicrystals and their Approximants in 2D Ternary Oxides, *physica status solidi (b)* 257, 1900624 (2020).

-Catalina Ruano-Merchan et al., Stoichiometry-Driven Formation of Two-Dimensional Ternary Oxides: From Quasicrystal Approximants to Honeycomb Lattice Structures, *J. Phys. Chem. C* 128, 8839 (2024).

Response to comment 3: The 5 side rings appear in the transverse section of the nanorods which may be considered by reviewer 4 as evidence for quasicrystal. However, we need to make it clear that **Fig.3c** in our manuscript is a STEM image of $\text{Nb}_{18}\text{W}_{16}\text{O}_{93}$ crystalline powder at its [001] projection (the target material for the PLD fabrication). As shown in **Figure R17**, we found the lower magnification STEM image (including Fig.3c in the image) of the $\text{Nb}_{18}\text{W}_{16}\text{O}_{93}$ crystalline at its [001] projection. The STEM image and its FFT show that the structure is $\text{Nb}_{18}\text{W}_{16}\text{O}_{93}$ crystalline (Schematic of the structure are shown in **Figure R16f.**) with point and line defects.

Figure R17 a STEM image of crystal $\text{Nb}_{18}\text{W}_{16}\text{O}_{93}$ (Blue square is **Fig.3c**), b FFT analysis of a, c SAED results of crystal $\text{Nb}_{18}\text{W}_{16}\text{O}_{93}$.

The basic units of a quasicrystal need to be ordered based on the reference mentioned by reviewer 4. (*Physical Properties of Quasicrystals*, 1999, Pages 51-89; *Physica Status Solidi (b)*, 2020, 257, 1900624; *Journal Physical Chemistry C*, 2024, 128, 8839) Same units only point towards the ordered directions. Thus, the order parameter for the quasicrystal units should be 1. We used a reference provided by reviewer 4, to calculate the order parameter of one of the quasicrystals presented in the reference (*Physica Status Solidi (b)*, 2020, 257, 1900624). The data collection process is shown in **Figure R18** and **Table 2**. The structure has a C12 symmetry. Thus, the angle θ of the basic units in quasicrystal needs to be processed by the following equation:

$$\theta_{quasi}^1 = \theta_{quasi} - \frac{n\pi}{6} \left(n \in \mathbb{Z}, 0 \leq \theta_{quasi}^1 \leq \frac{\pi}{6} \right)$$

The order parameter of the basic units in this quasicrystal can be calculated by the following equation:

$$S_{quasi} = \max(S_{quasi}(\alpha)) = \max\left(\left|\frac{\sum_{m=1}^m \cos(12(\theta_{quasi}^1 - \alpha))}{m}\right|\right) \alpha \in \left[0, \frac{\pi}{6}\right)$$

Using the above method, we can obtain the order parameters of different basic units (square, rhombus and regular triangle), which is 1. However, we also calculated the order parameters of the basic units ((3,0), (4,0), and (5,0/1) rings) of the 2D amorphous plane of Nb-W-O nanorods. The results show that the order parameters of all the basic units are quite small, as shown in **Table 1**. Thus, we can conclude that the structure of the 2D amorphous plane of Nb-W-O nanorods is not a 2D quasicrystal structure but a 2D amorphous structure with a small proportion of medium-range order in it.

[Figure Redacted]

Figure R18 a Quasicrystal schematic (*Physica Status Solidi (b)*, 2020, 257, 1900624), **b-d** Order parameter analysis on the basic units in quasicrystal.

Table 2 Angle distribution of the basic units in quasicrystal

	0°	30°	60°	90°	120°	150°
Square	16	0	17	0	19	0
Rhombus	2	4	6	4	2	3
Regular triangle	39	36	37	35	0	0

Comment 4: Therefore, I cannot accept this manuscript for publication in its current form. The authors need to identify precisely how their new observations and models extend the existing literature of semi-ordered ternary oxides.

Response to comment 4: The semi-ordered ternary oxides in the reference mentioned by reviewer 4, either have a crystal structure (same as our **Fig.3c**, the crystal structure of $\text{Nb}_{18}\text{W}_{16}\text{O}_{93}$), or exhibit order (2D quasicrystal structure). 2D amorphous plane of Nb-W-O nanorods does not have any of either. It can only be a 2D continuous random network amorphous plane with a small proportion of medium-range order in it.

Minor comments

Comment 5: “Scanning electron microscopy (SEM) results reveal 2, 1 and 3 orientational growth of Nb-W-O nanorods on Nb-doped SrTiO₃ (001), (011) and (111) substrates, respectively.” This statement is very unclear – what does “2 orientational growth” etc. mean? While I think I see what the authors are getting at, they should clarify that they mean families of nanowire orientation.

Response to comment 5: We have revised the sentence into: “Scanning electron microscopy (SEM) results reveal that Nb-W-O nanorods on Nb-doped SrTiO₃ (001), (011) and (111) substrates grow in 2, 1, and 3 orientations, respectively.”

Comment 6: Figure 4 caption is way too long. All the captions are too long, but especially this one. Concisely describe your observations and move discussion to the main text.

Response to comment 6: We revised the 3D RSM part into 2 Figures, as shown in **Figure R5** and **Figure R6**.

Comment 7: “... defined by the choice of substrate, rather than that it is the result of an uncontrolled local phenomenon.” Could be clarified.

Response to comment 7: We deleted this description as we found that the description is not relevant.

Response to Reviewer #5

General Comment: The authors have fabricated a unique three-dimensional structure composed of stacked two-dimensional amorphous layers in the Nb-W-O system. By taking advantage of the unidirectional periodicity of this structure, they performed direct imaging of the amorphous structure using state-of-the-art STEM techniques. This approach, combined with X-ray diffraction, EXAFS, and RSM analyses, enables a detailed structural characterization. A key strength of the study is that the electron beam was aligned parallel to the periodic axis, allowing unambiguous imaging of the amorphous structures. The experimental work is thorough, and the discovery of a 3D structure possessing 2D amorphous character is noteworthy. However, I have several concerns as listed below:

Response to General Comment: We thank Reviewer 5 for the insightful comments. We have revised our manuscript accordingly based on your valuable comments. By addressing these issues and incorporating the reviewer's feedback, we believe that our manuscript is now significantly improved.

Comment 1: The authors should address the relationship between their findings and previous work, particularly Ref. 8 (Huang et al., Nano Lett. 12, 1081 (2012)), where a two-dimensional amorphous SiO₂ film was directly imaged using STEM and shown to reproduce the classic Zachariasen model. Although the current study focuses on a three-dimensional structure, the amorphous nature is still essentially two-dimensional. The authors should clearly articulate the scientific advancement beyond previous studies.

Response to comment 1: We agree with reviewer 5 that the amorphous nature of Nb-W-O is 2D amorphous plane and our analysis approach is highly inspired by ref.8 in the manuscript. The 2D amorphous SiO₂ discovered by Huang et al. is the thinnest glass which is included in latest Guinness Book of World Records. However, the scientific advancement of our work is the first homogeneous material which shows a three-dimensional structure composed of crystalline stacked two-dimensional amorphous layers.

We revised the introduction as follows: "By taking advantage of the 2-dimensional nature of some materials, single-layer amorphous materials, since the first discovery by Huang⁸⁻⁹ as the thinnest glass, have expanded greatly including 2D amorphous carbon^{5,10}."

Comment 2: In the introduction, the authors refer to the longstanding debate on whether three-dimensional amorphous materials exhibit medium-range order (MRO). However, the present structure is fundamentally a two-dimensional amorphous material, and thus may not directly address this question. The authors are encouraged to comment on this point and clarify their position.

Response to comment 2: We provided a case which combines amorphous information (2D amorphous plane) and crystal information (crystal third direction) in a homogeneous structure at different directions, which suggests that “degree of order difference at different orientation” can be as important as “degree of order” itself for discussing a structure between complete amorphous phase to complete crystal phase.

We revised the introduction as follows: “The finding of such unique structure proposes a new insight for the boundary between amorphous and crystalline phases beyond the debate of medium-range order.”

Comment 3: The manuscript claims that the amorphous layers exhibit only short-range order (SRO), without MRO. Is this conclusion fully justified? For example, the reduced pair distribution function $G(r)$ shown in Fig. 2f extends only up to 5 angstrom, making it difficult to conclusively determine the absence of MRO. Furthermore, the peak near 4.8 angstrom might correspond to in-plane amorphous features and could be indicative of MRO, especially since it exceeds typical interatomic distances.

Response to comment 3: We admit that the unavoidable signal from SrTiO_3 in the PDF results due to the larger beam size compared with the thickness of the nanorods. After in depth analysis (inspired by reviewer 2) on the STEM image of the 2D amorphous plane of Nb-W-O nanorods, we found a small proportion (less than 5%) of MRO in it. The determined MRO structure was not originally in the $\text{Nb}_{18}\text{W}_{16}\text{O}_{93}$ lattice, which is the reason we didn't count it as medium-range order in the Nb-W-O system. Furthermore, the small proportion of MRO structure (4-side ring clusters) is evidence of the ring rearrangement from the original $\text{Nb}_{18}\text{W}_{16}\text{O}_{93}$ crystal.

Thus, we revised the description as follows:

“After detailed analysis (Supplementary Note 9, Figs. S42) on the STEM images of 2D amorphous layers of Nb-W-O nanorods (Fig.3 a-b), we can conclude that MRO exists in the 2D amorphous plane of Nb-W-O and the randomness of the 2D amorphous plane on SrTiO_3 (011) is less than the one on SrTiO_3 (001). However, both MRO area proportions are small and considering the small order parameter, we can still conclude that the 2D amorphous plane of Nb-W-O nanorods is a 2D amorphous plane with

continuous random network structure^{5,8-9} and the randomness of the plane would be similar when we use the same growth condition even with different oriented SrTiO₃ substrate.”.

However, small proportion of MRO can be found in most reported continuous random network structures (*Nano Letter*, 2012, 12, 1081-1086; *Nature*, 2020, 577, 199–203), as shown in **Figure R12-14**. Thus, we believe we can still define the 2D amorphous plane of Nb-W-O nanorods as a continuous random network 2D amorphous layer.

Comment 4: In Figs. 3a and 3b, STEM images of samples grown on SrTiO₃ (001) and (011) substrates are shown, respectively. Fig. 3b appears more ordered than Fig. 3a. Could this reflect a difference in the degree of medium-range order? The authors should address this observation and provide their interpretation.

Response to comment 4: Inspired by the comment from reviewer 2, we analyzed the MRO area of 2D amorphous planes of Nb-W-O nanorods on SrTiO₃ (001) and (011) (**Fig.2 a-b**), as **shown in Supplementary Note 9**. The MRO area proportion of 2D amorphous planes of Nb-W-O nanorods on SrTiO₃ (011) is a bit higher than the one on SrTiO₃ (001), 3.9% vs. 1.1%.

Furthermore, inspired by the order parameter definition of liquid crystal, we calculated the order parameter of the basic units (different rings) inside the 2D amorphous plane of Nb-W-O nanorods, as shown in **Supplementary Note 8**. The results of the order parameter analysis show that the degree of order of basic units in both 2D amorphous planes of Nb-W-O nanorods on SrTiO₃ (001) and (011) is small. And we cannot find a higher degree of order between 2D amorphous plane of Nb-W-O nanorods on SrTiO₃ (001) and the one on SrTiO₃ (011) by comparing the order parameter due to the inconsistency of the comparison of the order parameters in both cases. Thus, advanced models for analyzing degree of order along different orientations are urgently needed. We plan to analyze this in future studies.

Comment 5: It is possible to extract atomic coordinates from the bright spots in the STEM images and calculate the reduced pair distribution function $G(r)$. I recommend comparing this with the $G(r)$ obtained from XRD measurements. This comparison would help disentangle the contributions from amorphous in-plane structure and periodicity along the stacking direction.

Response to comment 5: We have used the model “ePDFpy” (*Computer Physics Communications*, 2024, 299, 109137) to transfer the SAED results of the amorphous

materials to PDF results. However, the thickness of the Nb-W-O nanorods films is smaller than the electron beam width for SAED analysis. Thus, we could not obtain a reliable SEAD result. However, we already listed some areas with MRO in the 2D amorphous plane and compared them with other reported 2D continuous random network amorphous structures. Thus, we believe we can conclude that a low contribution MRO in 2D continuous random network amorphous structure is not avoidable. We revised the discussion about the existence of “MRO” in our manuscript and we focused on the newly identified structure, and its contributions to the existing knowledge on the state of solid matter.

Comment 6: It remains unclear what conceptual or structural insights this study offers beyond the compatibility with Zachariassen's model. The authors are encouraged to more clearly state the broader implications of their findings for the structural science of amorphous materials, particularly in the summary section.

Response to comment 6: As discussed above, we present a material that is amorphous in the plane perpendicular to the nanorod, while being completely ordered, and therefore crystalline, along the length of the nanorod (**Fig.1 a-c**). This periodicity along the third-dimensional axis enabled the direct imaging of the amorphous 2D Nb-W-O monolayers, which are found to follow Zachariassen's model.

We believe that the contribution of our case is to provide new insight to distinguish the boundary between amorphous and crystalline cases, for which the direction matters. This is beyond current understanding in the debate how medium range disorder determines the distinct boundary between amorphous and crystalline materials.

We revised the summary section as follows: “Our work sheds light on the intricate interplay between order and disorder in solid matter and contributes to the understanding of the structural properties of amorphous/crystalline systems beyond the debate of medium-range order. As the new material bridges the gap between amorphous and crystalline matter, its unique properties will open new avenues for future exploration.”

Response to Reviewer #1

Overall Comment: This second version of the manuscript greatly improves the understanding of what the authors would demonstrate. After reading the new version, I believe the manuscript can be accepted for publication if the authors address the following two comments.

Comment 1: According to one of my comment on the previous version, the authors have now defined a clear orientation notation with respect to both the measurement orientation and the crystallographic main directions. Unfortunately, they used the symbol “*” to label the [100], [010] and [001] directions. This is a pity because this symbol is used in crystallography to represent directions in the reciprocal space. Another choice has to be done.

Response to Comment 1: We thank the reviewer for the constructive suggestion. We have changed x^* , y^* , z^* to x' , y' , z' in the real space to avoid misunderstanding.

We have revised the corresponding parts in the revised main text/figures and **Supplementary Note 1** to clarify those misunderstandings.

Comment 2: On page 2, the authors wrote that the material under study is amorphous in the x-y plane and this statement refers to the structural model reported Fig. 1 a-c. In fact, not any representation of structure in x-y plane is reported Fig. 1. The authors state that, due to the periodicity in the third direction, such an in plane structure cannot be represented. Nevertheless, in principle, a slice of the structure with a thickness of one atomic plane could evidence this in plane amorphous nature. This must be added.

Response to Comment 2: We thank the reviewer for bringing this to our attention, which has been modified in the illustration for better understanding. The single layer illustration is shown in **Figure R1**.

Figure R1 Single layer illustration from the Nb-W-O nanorods. **a** 2D view, **b** Transverse section, **c** Longitudinal section.

We have replaced **Figure 1b** with **Figure R1b** for a better understanding. The new **Figure 1** is shown below as **Figure R2**.

Figure R2 (Fig. 1) Schematics and SEM images of different type Nb-W-O nanorod films, **a** 3D Structure of Nb-W-O nanorods (z axis is the growth direction), **b** Transverse section (single atomic layer) of Nb-W-O nanorods, **c** Longitudinal section of Nb-W-O nanorods, **d** Schematic of the Nb-W-O nanorods on SrTiO₃ (001) substrate, **e-f** Top and side view SEM images of the Nb-W-O nanorods on SrTiO₃ (001) substrate, **g** Schematic of the Nb-W-O nanorods on SrTiO₃ (011) substrate, **h-i** Top and side view SEM images of the Nb-W-O nanorods on SrTiO₃ (011) substrate, **j** Schematic of the Nb-W-O nanorods on SrTiO₃ (111) substrate, **k-l** Top and side view SEM images of the Nb-W-O nanorods on SrTiO₃ (111) substrate. (In this work, x' , y' , z' are defined as the [100], [010], [001] directions of SrTiO₃ crystal, details are described in **Supplementary Note 1**.)

Response to Reviewer #2

Overall Comment: The reviewer has carefully checked the response to the comments on the manuscript entitled “Crystalline or Amorphous? A Matter of Perspective” by Xia et al. Although most questions have been properly addressed, the reviewer is unsatisfied with the response to comment 4. In the response to comments 4, the author wrote “The amount of Nb-W-O nanorods is negligible compared to the substrate. Thus, it is impossible to detect the Nb-W-O signal in differential scanning calorimetry. Furthermore, collecting a large enough amount of nanorods from the substrate surface to enable differential scanning calorimetry seems extremely challenging”. In fact, this reply is not true for flash DSC. The desired sample amount for flash DSC is only a few micrograms, and the Nb-W-O signal can be detected by scraping a few pieces of nanorods from the substrate. To determine the nature of this non-crystalline matter, the reviewer still recommends that the authors perform thermal measurements using flash DSC. However, this can be addressed in a future study. The manuscript has been significantly improved and it deserves the publication in Nature Communications.

Response to general comment: We thank the reviewer for pointing out the capabilities of flash differential scanning calorimetry, which is unfortunately not available within our institute. Furthermore, peeling off the Nb-W-O nanorods from an epitaxial substrate will be challenging and future studies can explore the usage of freestanding membranes (*Science*, 2024, **383**, 388-394).

Response to Reviewer #3

Overall Comment: I am still not entirely convinced about the author's claims of <5% medium range order (MRO) or the novelty of this study. By eye in Figures 1 and 2 I think I see correlations well beyond 5 nm, and I agree with the other referee that the PDF should be extended to longer length scales. By taking these image and looping through all (masked) bright sites, the authors could easily compute an imaging PDF / reduced PDF.

However I feel the authors have done their due diligence to investigate most of questions of myself and the other referees. Therefore I am prepared to accept this manuscript for publication, if the authors can fulfill this request:

-Please upload all of the raw TEM images published in this study to a public repository (such as Zenodo). It's not acceptable to write "Data Availability - The data that support the findings of this study are available from the corresponding author upon reasonable request." anymore and Springer Nature policy requires upload of all data (and analysis codes), here is the policy:

<https://www.nature.com/nature-portfolio/editorial-policies/reporting-standards#availability-of-data>

If the authors won't measure the long range PDF themselves, they must make it possible for the research community to do so (and I will be checking your results after publication!).

Response to general comment: We thank the reviewer for bringing this issue to our attention. We have uploaded the data and images supporting the findings of this study at Zenodo repository : <https://doi.org/10.5281/zenodo.17588010> , which will be made publicly available before publication of our manuscript.

We transferred the STEM images into PDF results, as shown in Figures R3, R4, and R5. Supplementary Note 5 and Figures S8-S10 have been added to describe obtained PDF results.

The STEM images of the transverse section of the as-grown Nb-W-O nanorods on SrTiO₃ (001) and (011) substrates as well as the (001) plane of crystalline Nb₁₈W₁₆O₉₃ powder were used to simulate the PDF results, which are shown in **Figures R3** and **R4**. As shown in **Figure R3**, only metal ions are clearly displayed in the STEM images. Thus, the PDF results calculated from STEM images would only reveal the statistics of the metal atom distances. As shown in **Figure R4**, the PDF result of crystalline Nb₁₈W₁₆O₉₃ shows a concentrated, periodic distribution of the metal atom distance. However, the PDF results of the transverse section of the as-grown Nb-W-O nanorods on SrTiO₃ (001) and (011) substrates show a constant G(r) at 1 after 1 nm atom distance, which provides solid evidence that the transverse section of the as-grown Nb-W-O nanorods on SrTiO₃ (001) and (011) substrates exhibits a 2D amorphous structure.

The following sentences have been added to the revised manuscript: *“The PDF results simulated from the STEM images are shown in Supplementary Note 5, Fig. S8-10 and only reveal the distance distribution between the metal ions. The conventional Nb₁₈W₁₆O₉₃ crystal exhibits multiple peaks distributed over distances till 60 Å, while in strong contrast the Nb-W-O nanorods on SrTiO₃ (001) and (011) substrates show relatively flat lines after 10 Å. These results confirm the 2D amorphous nature of the transverse section of Nb-W-O nanorods.”*

Figure R3 STEM image of (a, b) the transverse section of the as-grown Nb-W-O nanorods on SrTiO₃ (001) and (011) substrates as well as (c) the (001) plane of crystalline Nb₁₈W₁₆O₉₃ powder. d-f the atom searching results.

Figure R4 PDF results of the STEM images in Figure R3 obtained by the atom location data.

Figure R5 PDF results comparison between the simulated result from STEM image and tested result from synchrotron XRD of Nb-W-O nanorods on SrTiO₃ (011) substrate.

As shown in **Figure R5**, the smallest distance between the metal ions in the STEM image of Nb-W-O nanorods on SrTiO₃ (011) substrate is between 3.5 to 4.1 \AA , which spans 2 peaks in the synchrotron XRD result of the same sample. Thus, even the M-O bond and the signal from SrTiO₃ appeared in the synchrotron XRD data, next to the M-M bonds from the Nb-W-O nanorods.

Response to Reviewer #4

Overall Comment: I appreciate the authors' revisions in response to my comments. In particular, the modifications regarding Comments 1, 2, 3, and 4 satisfactorily address my concerns. However, I still have remaining concerns regarding my previous Comments 5 and 6.

Comment-1. With respect to my previous comment 5, thank you for the clarification regarding the difficulty of obtaining reliable SAED data for PDF analysis. However, I would like to emphasize that the main point of my earlier comment was not about deriving PDF from SAED, but rather about the possibility of extracting atomic coordinates directly from the STEM images and calculating the 2D reduced pair distribution function($G(r)$) from them. This approach does not rely on SAED and would make better use of the STEM data that the authors already possess. By comparing the STEM-derived $G(r)$ with the one obtained from XRD, the authors could provide a more quantitative basis for their discussion of medium-range order (MRO) and disentangle the respective contributions from in-plane amorphous structure and stacking periodicity. Such an analysis would significantly strengthen the manuscript and support the conclusions in a more rigorous way.

Response to Comment 1:

We transferred the STEM image into PDF results, as shown in Figure R3, R4 and R5. Supplementary Note 5 and Figures S8-S10 have been added to describe the obtained PDF results. See the description above answering the same comment by reviewer 3.

Comment-2. With respect to my previous comment 6, I appreciate the authors' efforts to revise the summary section. However, the response still does not fully address the essence of my concern. The main point was to clarify what conceptual or structural insights this study provides for the field of amorphous materials beyond demonstrating compatibility with Zachariassen's model. The revision largely restates the anisotropic order-disorder character of the nanorods, but it remains unclear how this observation advances broader understanding. In particular, the authors are encouraged to state more explicitly what the key factor is in distinguishing the boundary between amorphous and crystalline states, and what implications this may have for future studies of amorphous/crystalline systems.

Response to comment 2: Many thanks for raising this point. Distinguishing the boundary between amorphous and crystalline states is important because it helps us understand how the atomic structure of a material influences its physical, chemical, and functional properties. Determining this boundary is crucial for interpreting how structure governs behavior in materials. Understanding where and how this boundary lies enables better control of crystallization and amorphization processes, which are fundamental to materials synthesis, thin-film deposition, and annealing. Many emerging materials rely on properties that lie between amorphous and crystalline regimes. Understanding this boundary helps tune performance for desired applications. Our work opens important new avenues of exploration and provides a new platform for computational modeling of the 2D amorphous matter stacking. Moreover, accurate determination of amorphous solids in the complete range of disorder is

important to shed new insights in modelling of amorphous/crystalline systems in three-dimensional real space.

We have added this additional description to the summary section.

The manuscript entitled “Crystalline or Amorphous? A Matter of Perspective” by Xia *et al.* demonstrates an interesting inorganic matter which is amorphous in two dimensions but crystalline in the third. The crystalline (including quasicrystal) and amorphous structures are two states for solid matters. They are commonly believed to be mutually exclusive that a state of matter cannot be both crystalline and amorphous at the same time. Usually, a partially crystalline or amorphous matter is just a composite of purely crystalline and amorphous parts. Hence, the matter demonstrated in this manuscript presents experimentally a novel state which is only ordered in certain dimensions.

In fact, such a state of matter has been theoretically proposed as paracrystalline. The term paracrystal refers to a state of matter with less than three-dimensional order, characteristic of a true crystal (Ref: <https://www.merriam-webster.com/dictionary/paracrystal>). Paracrystals are often used in materials science to understand the transition between crystalline and amorphous states. Although the paracrystalline matter has been demonstrated microscopically in many papers (*Science* 335, 950, (2012); *Nature* 599, 605 (2021); *JAP* 90, 4438 (2001)), a direct observation of such matter in bulk is still missing. In this manuscript, we can see the micrometer-scale formation of paracrystalline matter, and the structural characterizations provide compelling evidence in a degree of intermediate order between crystalline and amorphous materials. This manuscript contributes significantly to the existing knowledge on the state of solid matter.

Reviewer has carefully examined the manuscript and supports its publication in *Nature Communications*. However, it will be shown in the following comments why this matter should

be considered as paracrystalline rather than amorphous in one direction.

(1) In Fig. 1, Nb-W-O nanorod films were deposited on crystalline substrates of different orientations. It is a smart approach to demonstrate the difference in structures between different orientations of a matter by showing the various growth morphologies. However, why do these rods all have a diameter around 50 nanometers? If it is truly amorphous in the x-y plane, the diameters of these rods should be infinite or distributed randomly.

(2) What would happen if the films were deposited on an amorphous substrate?

(3) In Fig. 2a and b, the author tries to prove the amorphous structure of the x-y plane by showing its STEM image and the corresponding FFT pattern. However, when the review carefully checks them, some tiny crystalline features can be observed as shown in the following Figures 1R, 2R and 3R. Hence, the x-y plane cannot be understood as fully amorphous but paracrystalline phase.

Fig. 1R. (a) STEM image of Fig. 2a in manuscript and (b) its corresponding halo FFT pattern. If we tuned the intensity of the FFT pattern, some diffraction spots corresponding to crystalline phase

can be observed.

Fig. 2R. If we zoom in Figure 2a, a lot of tiny crystals can be observed.

Fig. 3R. Another region where the tiny crystals can be observed. Such regions are distributed over the whole image.

- (4) The most important feature of amorphous matter differing from crystalline matter is the glass transition. Can the glass transition be observed in this matter by differential scanning calorimetry? If not, then this matter is paracrystalline rather than amorphous.

By the way, some typos can be seen in the manuscript. “F PDF of Nb-W-O nanorods”.

In summary, the state of matter in this manuscript is still unclear. The author needs to verify which kind of structure it is, amorphous or paracrystalline?